# Daclizumab: Mechanisms of Action, Therapeutic Efficacy, Adverse Events and Its Uncovering the Potential Role of Innate Immune System Recruitment as a Treatment Strategy for Relapsing Multiple Sclerosis

**DOI:** 10.3390/biomedicines7010018

**Published:** 2019-03-11

**Authors:** Stanley L. Cohan, Elisabeth B. Lucassen, Meghan C. Romba, Stefanie N. Linch

**Affiliations:** 1Providence Multiple Sclerosis Center, Providence Brain and Spine Institute, Portland, OR 97225, USA; elisabeth.lucassen@providence.org (E.B.L.); meghan.romba@providence.org (M.C.R.); 2Providence Health and Services, Regional Research Department, Portland, OR 97213, USA; stefanie.linch@providence.org

**Keywords:** daclizumab, relapsing multiple sclerosis, CD25, innate immune system, interleukin-2, drug reaction with eosinophilia systemic symptoms, DRESS, autoimmunity

## Abstract

Daclizumab (DAC) is a humanized, monoclonal antibody that blocks CD25, a critical element of the high-affinity interleukin-2 receptor (IL-2R). DAC HYP blockade of CD25 inhibits effector T cell activation, regulatory T cell expansion and survival, and activation-induced T-cell apoptosis. Because CD25 blockade reduces IL-2 consumption by effector T cells, it increases IL-2 bioavailability allowing for greater interaction with the intermediate-affinity IL-2R, and therefore drives the expansion of CD56^bright^ natural killer (NK) cells. Furthermore, there appears to be a direct correlation between CD56^bright^ NK cell expansion and DAC HYP efficacy in reducing relapses and MRI evidence of disease activity in patients with RMS in phase II and phase III double-blind, placebo- and active comparator-controlled trials. Therapeutic efficacy was maintained during open-label extension studies. However, treatment was associated with an increased risk of rare adverse events, including cutaneous inflammation, autoimmune hepatitis, central nervous system Drug Reaction with Eosinophilia Systemic Symptoms (DRESS) syndrome, and autoimmune Glial Fibrillary Acidic Protein (GFAP) alpha immunoglobulin-associated encephalitis. As a result, DAC HYP was removed from clinical use in 2018. The lingering importance of DAC is that its use led to a deeper understanding of the underappreciated role of innate immunity in the potential treatment of autoimmune disease.

## 1. Introduction

Daclizumab (DAC) is a humanized monoclonal antibody that blocks CD25, the α-subunit of the high-affinity interleukin-2 receptor (IL-2R-HA). DAC was initially developed as an intravenous therapy (Zenapax, Hoffman-LaRoche, Nutley, New Jersey) to prevent transplant organ rejection and to treat T cell leukemia and severe uveitis [1,2,3,4]. Subcutaneous DAC, often referred to as DAC High Yield Process (DAC HYP; Zinbryta, Biogen, Cambridge MA and AbbVie, Chicago II), demonstrated efficacy in reducing both clinical and MRI measures of disease activity in patients with relapsing forms of multiple sclerosis (RMS) and was approved to treat RMS in 2016 [5,6,7,8]. Unfortunately, secondary autoimmune disease directed primarily against the central nervous system (CNS), liver, and skin resulted in serious adverse events (SAE) leading to its withdrawal in 2018. However, investigations of its mechanisms of therapeutic action and immunomodulatory activity have revealed insights into heretofore underappreciated relationships between innate and adaptive immunity in autoimmune diseases, particularly RMS. Future exploitation of these relationships may provide further therapeutic opportunities. This review will discuss known and proposed, direct and indirect effects of DAC HYP upon the immune system and how these effects are potentially related to its therapeutic impact in RMS. To better appreciate the multiple mechanisms by which DAC may impact autoimmunity, a brief description of the major immune system components that it is believed to affect, and the downstream impact of those actions, will be reviewed.

## 2. Immunologic Background for DAC Efficacy

### 2.1. IL-2 and IL-2R Signaling

IL-2 is a trophic cytokine expressed predominantly by activated T cells, which occurs upon cognate antigen recognition by the T cell receptor. While IL-2 promotes proliferation and survival of both activated effector and naïve T cells, it also promotes the expansion of regulatory T cells (T_reg_ cells), and some authors have reported that IL-2 activation may also promote apoptosis of antigen-activated T cells, thereby contributing to the maintenance of immune homeostasis [9,10,11,12,13,14,15]. Although most IL-2 is produced by activated T cells, it is also produced by myeloid dendritic cells (mDC) and macrophages [16,17].

IL-2 activity is primarily mediated through binding to the IL-2 high-affinity receptor (IL-2R-HA), which, in addition to IL-2, is comprised of three subunits: the α-subunit (CD25), the β-subunit (CD122), and the common γ-chain subunit (CD132), as shown in Figure 1. CD122 and CD132 can also combine to form an intermediate-affinity IL-2R (IL-2R-IA). CD132 is expressed on all lymphoid cells, whereas CD122 is constitutively expressed on natural killer (NK) and CD8 memory T cells, and can be induced on naïve T cells following antigen recognition. CD25 expression is much more tightly restricted; it is absent on naïve and memory cells, but is induced following antigen activation [17,18]. Furthermore, CD25 is constitutively expressed at very high levels on T_reg_ cells and is vital for the development and peripheral maintenance of the T_reg_ cell population [19]. It is also expressed on mDC, which may assist in T cell activation by using CD25 to “trans-present” IL-2 to a CD25-deficient IL-2R expressed on recently activated T cells, thus converting it to the IL-2R-HA [20].

Although both CD122 and CD132 are constitutively expressed on lymphocytes, IL-2 must be bound to CD25 to recruit these subunits to form the IL-2R-HA complex [21,22]. While IL-2 can also bind and signal through the IL-2R-IA, receptor affinity increases 1000-fold following its association with CD25 [18]. Furthermore, in the absence of CD25, the IL-2R-IA requires up to 50 times the IL-2 available at basal physiological states [23]. IL-2 bound to the IL-2R-HA results in rapid internalization and degradation of the high-affinity receptor complex. This latter step, terminating IL-2-induced T cell activation via the intracellular domain of CD122, may be initiated by activation of the intracellular domain of the CD132 portion of the IL-2R, thereby regulating IL-2 effects [24,25,26]. Subsequently, CD25 can be recycled back to the surface of the cell for future engagement with CD122, CD132, and IL-2 [27,28].

### 2.2. Innate Lymphoid Cells (ILC)

Innate lymphoid cells (ILC) arise from a hematopoietic CD34^+^ cell line, which is considered the common progenitor for all ILC [29]. Multiple subsets of ILC have been identified, of which several are relevant to the potential immunoregulatory actions of DAC: group 1, which includes ILC1 and CD56^bright^ NK cells; group 2, comprised of ILC2, to which we will return later; and group 3, which includes ILC3 and pro-inflammatory lymphoid tissue inducer (LTi) cells [30,31,32,33]. NK cells constitutively express CD122 and CD132. Notably, CD56^bright^ NK cells have 10-fold greater expression of CD122 relative to the pro-inflammatory CD56^dim^ NK cells [34]. IL-2 stimulation of the IL-2R-IA has been reported to drive ILC toward anti-inflammatory CD56^bright^ NK cell production, possibly at the expense of the pro-inflammatory LTi cell pathway [11,35,36].

There is considerable evidence to support a role for CD56^bright^ NK cells in auto-regulation and tolerance. CD56^bright^ NK cells can destroy activated immune cells through direct contact, including autologous antigen-activated CD4^+^ and CD8^+^ T cells, which play a critical role in autoimmune responses [34,37,38]. In support of this concept, reduced CD56^bright^ NK cell cytotoxicity was observed in patients with MS as early as the 1980s [39,40,41,42]. Subsequently, investigators demonstrated that CD56^bright^ NK cells are directly cytotoxic to auto-antigen-specific T cells [34,38]. Of note and expanded upon below, there is a strong correlation between DAC-induced up-regulation of CD56^bright^ NK cells and the therapeutic efficacy of DAC HYP in RMS [34,43,44].

## 3. DAC Mechanisms of Action: Known and Proposed

DAC reversibly binds to CD25, preventing the interaction of IL-2 with the IL-2R-HA, leaving more IL-2 available to induce IL-2R-IA signaling. As noted above, the IL-2R requires CD25 to render it high-affinity. By blocking the formation of IL-2R-HA, DAC blocks IL-2 consumption by activated effector T cells, causing a moderate decrease in the frequency of CD4^+^ and CD8^+^ T cells in the periphery and increasing the availability of IL-2 to interact with the IL-2R-IA [34,37,43,45]. As might be expected with a reduction in IL-2 consumption, IL-2 concentrations in the serum were elevated following DAC treatment by approximately 50% [46]. Because expansion of T_reg_ cells is dependent upon IL-2 interaction with the IL-2R-HA, as predicted, DAC treatment also resulted in a 50–60% decrease in peripheral T_reg_ cells [37,46,47,48], as shown in Figure 2. By blocking the IL-2R-HA, DAC may also potentially prevent activation-induced T cell apoptosis, paradoxically inhibiting auto-regulation by allowing the survival of activated effector T cells recognizing “self” antigens [10,11]. DAC HYP administration has also been reported to normalize previously elevated intrathecal CD4^+^, CD8^+^ and HLA-DR-expressing CD4^+^ T cells and B cells in patients with RMS [35]. In some patients, the effect of DAC on effector T and T_reg_ cells may not be adequately compensated for by CD56^bright^ NK cell expansion, as is observed in patients with spontaneously occurring autoimmunity due to genetic variants that delete CD25 [49,50]. These resultant autoimmunity-promoting circumstances may also explain, in part, the paradoxical increase risk of autoimmune adverse events (AE) observed in some DAC HYP-treated RMS patients (see below).

Although part of the anti-inflammatory activity of DAC is through selective inhibition of IL-2-driven effector T cell activation and expansion, it is evident that this does not entirely account for its therapeutic efficacy [37]. Innate immune components also play an essential role in auto-tolerance and autoimmunity. As mentioned above, CD56^bright^ NK cells from multiple sclerosis patients exhibit reduced cytotoxicity. This is important because CD56^bright^ NK cells are believed to be directly cytotoxic to auto-antigen activated T cells [34,38]. CD56^bright^ NK cells also express high levels of cell surface IL-2R-IA, and IL-2 activation of the IL-2R-IA through DAC resulted in CD56^bright^ NK cell expansion, which was abolished by IL-2 inhibition [37]. IL-2 may also enhance CD56^bright^ NK cell maturation and reduce NK cell death, and further increase NK cell cytotoxicity, through enhanced killing efficiency [37,51] as shown in Figure 2. In prospective clinical studies, DAC HYP treatment resulted in an expansion of CD56^bright^ NK cells in up to 90% of DAC HYP-treated patients, with a 5-fold elevation of CD56^bright^ NK cell counts seen by 52 weeks of treatment that was sustained thereafter [44]. By blocking consumption of IL-2 by the IL-2R-HA, IL-2 levels increase systemically, thereby increasing the bioavailability of IL-2 to interact with the IL-2R-IA and preferentially promoting the expansion of anti-inflammatory CD56^bright^ NK cells via CD122 intracellular signaling [52].

It has also been reported that DAC reduces the expansion of pro-inflammatory LTi, as shown in Figure 3 [34]. By orienting the innate CD34^+^ progenitor cell line away from the pro-inflammatory LTi cell line, DAC may indirectly reduce the formation or activity of meningeal lymphoid follicle-like structures, which have been observed overlying cortical areas of demyelination [53,54,55]. Consistent with this concept, DAC HYP administration reportedly normalized previously elevated LTi cell numbers in the spinal fluid of patients with RMS [35]. One study found that the total number of ILC was reduced in DAC HYP-treated patients; however, this remains controversial and has not been replicated by subsequent research [31,35,36,56,57].

Because CD25 is also expressed on mDC, which may use CD25 to “trans-present” IL-2 to CD25-deficient T cells, it is possible that may block mDC potentiation of IL-2R-HA signaling on effector T cells, as shown in Figure 2a [20]. There is also some preclinical evidence to suggest that the modulation of cytokine production by mDC may also be a target of DAC therapy [58]. Further research is needed to confirm these reported effects of DAC on mDC.

In summary, activated effector T cell expansion is dependent upon IL-2 interaction with IL-2R-HA expressed on the cell surface. To perform as a high-affinity receptor, the IL-2R requires CD25, which is blocked by DAC, reducing IL-2 utilization by T cells. This drives IL-2 towards IL-2R-IA and CD56^bright^ NK cell expansion, which destroy activated effector T cells by direct contact. As will become evident from the paragraphs that follow, there is a direct correlation between the DAC-induced decrease in activated effector T cells and CD56^bright^ NK cell expansion with both clinical and MRI efficacy of DAC. However, there is also a reduction in T_reg_ cells, and possibly a decrease in activated T cell apoptosis, both of which may paradoxically promote the up-regulation of heightened autoimmunity, leading to SAE which will be discussed further below.

## 4. Pharmacokinetics and Pharmacodynamics of DAC HYP

Initial studies of DAC HYP in healthy adult subjects (ages 18–65) showed that the maximum serum concentration (C_max_) was 7 days while the t^1/2^ of elimination was 24 days [59]. In a subsequent study in adults with RMS, C_max_ was achieved in 5–7 days, and the t^1/2^ of elimination was approximately 21 days after the last injection [60]. Maximum CD25 saturation by DAC HYP was reached by 7 h and was sustained as long as DAC HYP serum concentration was 5.0 mcg/mL or higher. Baseline CD25 saturation was achieved in approximately 24 weeks following DAC HYP discontinuation, in parallel with a serum DAC HYP concentration of 1.0 mcg/mL or less. During treatment with DAC HYP (150 mg every 4 weeks), maximum CD56^bright^ NK cell level is reached by approximately 36 weeks, achieving a maximum 5-fold increase over baseline [47]. For patients treated with 300 mg DAC HYP, the changes in NK cell and T_reg_ frequency were nearly identical; no other results were reported separately for this group [47].

Several other studies examined various immune parameters in pooled DAC HYP-treated patients (150 mg and 300 mg) from the SELECT, SELECTION, and DECIDE trials. Following DAC HYP cessation, in both 150 mg DAC HYP alone and pooled (150 mg and 300 mg) DAC HYP groups CD56^bright^ NK cells return to pre-treatment levels by 24 weeks after the last injection. In parallel with increased CD56^bright^ NK cells, serum IL-2 levels increased by roughly 50% within 4 weeks of DAC HYP treatment and T_reg_ cell counts fell by 50–60% within 4 days of a 150 mg DAC HYP injection and returned to pre-treatment levels approximately 20 weeks after the last DAC HYP injection [34,44,46,47]. CD4^+^ and CD8^+^ T lymphocyte counts were decreased by 7–10% after 52 weeks and by 15–18% after 96 weeks of DAC HYP treatment [7]. Importantly, there was also a roughly 25% decrease in HLA-DR2-expressing activated effector CD4^+^ T cells, in DAC HYP treatment in patients with RMS in both the 150 mg and 300 mg groups similarly [61]. This coincides with the timing of maximum inflammatory lesion reductions observed in the CHOICE study and provides further evidence of the down-regulation of the pro-inflammatory activity of DAC HYP [5].

## 5. DAC Efficacy: Controlled Clinical Trial Data

As noted in the section on mechanisms of action, DAC administration expands immunoregulatory CD56^bright^ natural killer cells, reduces whole blood and intrathecal pro-inflammatory T cells and T_reg_ cells. Based on its immunologic effects, subcutaneous use of DAC HYP was approved by the United States Food and Drug Administration for RMS in May 2016 and the European Medicines Agency (EMA) in July 2016 [62].

Several small open-label studies evaluating intravenous DAC in RMS demonstrated improvement in several outcome measures [43,63,64]. Due to cell-mediated toxicities reported with intravenous DAC preparations, it was subsequently reformulated to DAC HYP, facilitating subcutaneous administration [59]. CHOICE was a randomized phase II double-blind, placebo-controlled trial in which DAC HYP was an add-on therapy to interferon beta (IFNβ) versus placebo plus IFNβ. CHOICE demonstrated that DAC HYP administered at 2 mg/kg with IFNβ resulted in a 72% decrease in the number of contrast-enhancing lesions relative to IFNβ with placebo (*p* = 0.004) [5], as shown in Table 1.

DAC HYP was subsequently studied in two large, pivotal phase III clinical trials in patients with RMS; a 52-week placebo-controlled trial (SELECT) and an active comparator trial (DECIDE) of DAC HYP versus IFNβ. The primary endpoint of the SELECT and DECIDE trials was the annualized relapse rate (ARR). Secondary and tertiary outcomes measured include: impact on 3 month confirmed disability progression, T2 lesion burden and gadolinium-enhancing (Gd^+^) lesions, safety, measurement of markers of immune activity; and in DECIDE, changes in cognitive status and patient-reported outcomes [6,7,65].

### 5.1. SELECT

In SELECT, patients with RMS were randomized 1:1:1 to subcutaneous DAC HYP 150 mg, DAC HYP 300 mg, or placebo every four weeks for 52 weeks. The primary endpoint was met with a 54% relative reduction in ARR (95% CI 33–68%; *p* < 0.0001) in the group treated with DAC HYP 150 mg (ARR 0.21) and 50% reduction (95% CI 28–65%; *p* < 0.0001) in those treated with DAC HYP 300 mg (ARR 0.23) in comparison to placebo (ARR 0.46). At 52 weeks, the three-month sustained disability progression decreased by 57% in the DAC HYP 150 mg group and 43% in the DAC HYP 300 mg group compared to placebo [7], as shown in Table 1.

Subcutaneous DAC HYP also demonstrated effects on brain MRI measures in SELECT. There was a statistically significant relative decrease in the number of Gd^+^ lesions (*p* < 0.0001) at week 52, with a 69% reduction in the DAC HYP 150 mg group versus 78% reduction in the DAC HYP 300 mg group compared to placebo. The number of new/newly enlarging T2 hyperintense lesions were also significantly reduced at week 52 (*p* < 0.0001) in the 150 mg group by 70% versus 79% in the 300 mg group compared to placebo [7], as shown in Table 1. The mean percentage of brain volume loss was not significantly different between the treatment arms in SELECT. Exploratory studies performed in SELECT demonstrated a strong correlation between early increases in CD56^bright^NK cells and decrease in new and enlarging T2 lesions at 24 and 52 weeks of DAC HYP treatment, and DAC HYP patients in the highest CD56bright NK cell quartile had 62% fewer new and enlarging T2 lesions than those in the lowest quartile [44].

### 5.2. SELECTION

SELECTION was a double-blind extension of SELECT, designed to further assess the risks of AE associated with prolonged DAC HYP therapy (52 weeks), the impact of discontinuing DAC HYP, and to determine if early efficacy was sustained. Placebo-treated patients in SELECT were randomized 1:1 to initiate either 150 mg or 300 mg DAC HYP subcutaneously every four weeks (“Switch” in Table 1). The patients who received DAC HYP during SELECT either continued the same medication dose through SELECTION for a total of 104 weeks of continuous therapy, or underwent a blinded, placebo-treated washout that lasted for a total of 20 weeks, followed by resumption of their prior DAC HYP dose for the remaining 32 weeks of their participation in SELECTION [8]. Therapeutic efficacy achieved by DAC HYP therapy in SELECT was measured by: ARR, proportion of patients experiencing a relapse (PPRF), 12-week confirmed disability worsening, number of new Gd^+^ lesions, number of new and enlarging T2 lesions, volume of T2 lesions, mean and total T1 lesion volume was sustained by week 52 of SELECTION (104 weeks of total DAC HYP therapy).

For patients treated with placebo in SELECT, treatment with DAC HYP during SELECTION resulted in a statistically significant decrease in ARR, the proportion of patients having relapses, the number of Gd^+^ enhancing lesions, new and enlarging T2, and the volume of T2 lesions. No significant differences were observed in T1 lesion measurements or whole brain volume, as shown in Table 1. For DAC HYP-treated patients in SELECT who underwent the 20-week washout period in SELECTION, clinical and MRI end-points were similar after 52 weeks in SELECTION to those patients who received DAC HYP for the entire 52 weeks in SELECTION, as shown in Table 1. By the end of the washout period, CD56^bright^ NK counts had returned to baseline values, but at the conclusion of SELECTION (32 weeks of DAC HYP treatment), CD56^bright^ NK numbers were similar to those found in patients who had been continuously treated with DAC HYP through the 104 weeks of SELECT and SELECTION.

### 5.3. DECIDE

DECIDE was a double-blind and double-dummy, active-comparator, phase III study, and compared efficacy in patients with RMS, randomized 1:1 to either subcutaneous DAC HYP 150 mg every four weeks plus weekly intramuscular placebo, or to IFNβ (30 µg, intramuscularly) once weekly, and placebo subcutaneously every four weeks, for up to 144 weeks [6]. The primary end point was met with a 45% reduction in annualized relapse rate (ARR 0.22) in the DAC HYP treated group compared to the IFNβ group (ARR 0.32) *p* < 0.001. The DAC HYP group also demonstrated a significant reduction in the annualized rate of severe relapses (38% reduction, *p* = 0.002). The number of patients who remained relapse-free at week 144 was 67% in the DAC HYP group and 51% in the IFNβ group. Three-month confirmed disability progression measured at week 144 revealed a 16% relative risk reduction in the DAC HYP cohort compared to IFNβ, which did not reach statistical significance. However, the risk of 24-week confirmed disability worsening demonstrated DAC HYP significantly reduced disability worsening by 27% compared to IFNβ (*p* = 0.03)

There was a 75% relative decrease (*p* < 0.001) in new Gd^+^ lesions and a 54% relative reduction (*p* < 0.001) in the number of new or newly enlarged T2 hyperintense brain MRI lesions over a 96-week period in patients treated with DAC HYP compared to IFNβ (*p* < 0.001). There was also a statistically significant decrease in brain volume loss in DAC HYP versus IFNβ treated patients after two years of treatment (*p* < 0.01). No evidence of disease activity (NEDA) over 96 weeks, as measured by no relapses, no disability progression, no new/enlarged T2 or Gd^+^ lesions, was seen in 22% of DAC HYP and 13% of IFNβ-treated patients (*p* < 0.001).

Investigators also observed a 7–10% drop in total CD4^+^ and CD8^+^ T lymphocyte counts over 52 weeks in the DAC HYP treated group in SELECT versus 15–18% over 96 weeks among the DAC HYP treated group in the DECIDE trial. The ratio of CD4^+^ to CD8^+^ T cells remained stable in the DAC HYP group in both SELECT and DECIDE. No correlation was found between the reduced CD4^+^ and CD8^+^ T cells and the infection rate in the DECIDE trial [6].

A tertiary outcome in DECIDE was the assessment of cognitive outcomes, as measured by changes in the orally administered Symbol Digit Modalities Test (SDMT). At 96 weeks, there was a significantly greater (*p* = 0.0274) improvement from baseline SDMT in DAC HYP-treated patients compared to IFNβ treated patients [65]. Furthermore, a significantly greater number of DAC HYP-treated patients experienced clinically meaningful improvement (*p* = 0.0366) and a significantly smaller number of patients experienced a clinically-meaningful decline (*p* = 0.0103) in their SDMT scores.

### 5.4. SELECTED

SELECTED was a single-armed open-label extension trial of DAC HYP 150 mg, for which any patient completing SELECT and SELECTION qualified [66]. Although the primary outcomes dealt with AE (see below), adjusted ARR was calculated every six months from the time the patient received their first dose of DAC HYP dose in either SELECT or SELECTION up to 144 weeks; changes in the number of new/enlarging T2 lesions and brain volume (percentage) compared to baseline, were obtained on yearly MRI scans. ARR varied from 0.21 in patients having received DAC HYP for 0–24 weeks and gradually declined to 0.15 in patients receiving DAC HYP for 121–144 weeks. The adjusted mean number of new/enlarging T2 lesions was 1.96 after year 1, 1.62 at year 2 and 1.26 at year 3. The annualized median decrease in brain volume from the first dose of DAC HYP was 0.77 at year 1, 0.57 at year 2, and 0.32 at year 3. As in all single arm open-label studies, efficacy outcomes may have been affected by selection bias arising from dropout of poor responders, and subjects with AE.

### 5.5. Patient-Reported Outcomes

The DAC HYP treatment groups in both SELECT and DECIDE demonstrated significant benefits in patient-reported outcomes, when utilizing the 29-item Multiple Sclerosis Impact Scale (MSIS-29), which evaluates the impact of MS on physical (PHYS) and psychological health (psychological impact scale), as well as EuroQol 5-Dimensions (EQ-5D), which is used to examine overall health status using five domains. In SELECT, there was a significant improvement in MSIS-29 PHYS score at week 52 in the subcutaneous DAC HYP 150 mg compared with placebo (*p* = 0.00082), but not with the DAC HYP 300 mg group. There was also a statistically significant benefit for subcutaneous DAC HYP 150 mg for EQ-5D index (*p* = 0.0091) compared to placebo in SELECT.

In DECIDE, DAC HYP 150mg treated patients showed a greater improvement compared to IFNβ at 96 weeks in regards to MSIS-29 PHYS (*p* < 0.001), MSIS-29 psychological impact scale (*p* = 0.04), and EQ-5D index (*p* = 0.005). Clinically meaningful worsening was defined as an increase in ≥7.5 points in the patient-reported outcomes at 96 weeks. As measured by MSIS-29 physical subscale, worsening was 19% in the DAC HYP group versus 23% in the interferon IFNβ group (NS). DECIDE also demonstrated a greater improvement in Multiple Sclerosis Functional Composite (MSFC) at 96 weeks in patients treated with DAC HYP compared to IFNβ in DECIDE (*p* < 0.001) [6].

## 6. Adverse Events with DAC: Prospective Clinical Trial Data

### 6.1. SELECT

In the randomized, double-blind, placebo-controlled SELECT trial, SAE excluding relapse was similar between groups, as shown in Table 2. Nine (2%) patients treated with DAC HYP developed serious infections, one of whom discontinued treatment, while six resumed treatment after the resolution of infection. No placebo-treated patients had serious infections. The frequency of oral herpes and herpes zoster infections was similar in all groups.

Four malignancies were reported during the trial, including two cervical carcinomas (one in the placebo, one in DAC HYP 150 mg) and two melanomas in the DAC HYP 300 mg group. Two patients in the DAC HYP 150 mg group and three in the DAC HYP 300 mg group, had serious cutaneous events, including rash, atopic and allergic dermatitis, ex-foliative dermatitis, and erythema nodosum, as shown in Table 3. While recovering from a serious rash, one DAC HYP (150 mg) patient died due to a psoas abscess. No serious cutaneous events were seen in the placebo group. More patients in the DAC HYP groups had liver transaminase increases of greater than five times the upper limit of normal (>5× ULN). These increases had a median onset at 308 (±SD) treatment days. Continued DAC HYP dosing did not increase recovery time of transaminase abnormalities, and 7/17 patients with elevated alanine aminotransferase levels >5× ULN continued or resumed DAC HYP after enzyme recovery, and did not experience a recurrence of liver function test abnormalities in the five months that followed [7].

### 6.2. SELECTION

In the multicenter, randomized, double-blind SELECT extension trial (SELECTION), frequencies of AEs and SAEs were similar between the patients switched from placebo to DAC HYP treatment and the continuous DAC HYP treatment groups, as shown in Table 2. Infections were reported in 42% of patients; 3% were considered serious infections. The only serious infection occurring in more than one patient was bronchitis. All infections were resolved with standard-of-care.

Breast cancer was reported in one (<1%) patient, which investigators did not believe was treatment-related. Cutaneous events occurred in up to 24% of patients as shown in Table 3. Serious cutaneous events were reported in six (1%) patients. These events included drug eruption, eczema, pityriasis rubra pilaris, exfoliative dermatitis, and urticaria. Alanine or aspartate aminotransferase levels of >5× ULN were observed in 11 patients (2%) and resolved in a median of 84.5 days. Of these patients, 10 resumed DAC HYP without recurrence; one did not resume treatment. One patient died due to autoimmune hepatitis following re-initiation of 300 mg DAC HYP; this was confirmed by liver histology upon autopsy. The contribution of DAC HYP could not be ruled out [8].

### 6.3. DECIDE

In the double-blind, active-controlled, randomized phase III DECIDE trial comparing DAC HYP 150 mg with IFNβ weekly injections, the overall incidence of AEs was similar among all groups, leading to discontinuation in 14% of DAC HYP-treated patients and 9% in the IFNβ treated group, as shown in Table 2. SAEs (excluding relapse) occurred in 15% of the patients treated with DAC HYP and in 10% of those treated with IFNβ. One patient in the DAC HYP group and four in the IFNβ died, but none of these deaths was considered to be related to study treatment according to investigators.

Infections occurred in 65% of DAC HYP patients and 57% of IFNβ patients. Five DAC HYP patients (1%) and three (<1%) IFNβ patients discontinued treatment due to infection. Common infections included nasopharyngitis (25% in DAC HYP group; 21% in IFNβ group), upper respiratory infection (16% and 13%, respectively), and urinary tract infection (UTI; 10% and 11%, respectively). The incidence of herpes virus infections, including herpes zoster, was similar in both treatment groups. Serious infections were reported in 4% of patients in the DAC HYP group and in 2% of the IFNβ group, including UTI, cellulitis, appendicitis, pneumonia, and viral infection. There were no reported cases of progressive multifocal leukoencephalopathy or infectious encephalitis.

Malignancies were found in seven DAC HYP patients and eight IFNβ patients, and no deaths related to treatment [6]. Cutaneous events occurred in 37% of DAC HYP patients and 19% of IFNβ patients, leading to treatment discontinuation in 5% and 1% of patients, respectively, as shown in Table 3. The most frequent cutaneous events were rash (7% in DAC HYP and 3% in IFNβ) and eczema (4% and 1%, respectively). Serious cutaneous events were observed in 2% of DAC HYP patients and <1% of IFNβ patients, and included dermatitis and angioedema. Hepatic events occurred in 16% of DAC HYP- and 14% of IFNβ-treated patients (serious events in 1% and <1%, respectively). Transaminase elevations >5× ULN occurred in 6% of DAC HYP and 3% of IFNβ patients. Elevations of transaminase levels were more common during the first year of IFNβ treatment, but were distributed evenly throughout DAC HYP treatment.

### 6.4. SELECTED

In SELECTED, the open-label DAC HYP extension study, SAEs (not including MS relapse) occurred in 16%. Discontinuation of treatment due to AEs including MS relapse occurred in 12% of patients, as shown in Table 2. The most common AEs (not including MS relapse) included nasopharyngitis and upper respiratory infection (both 12%). The most common SAEs (not including MS relapse) included elevated hepatic enzymes, ulcerative colitis, pneumonia, and UTI (<1% for each). Infections were observed in 50% of patients; 3% were considered serious. The infection incidence did not increase over time, and less than 1% of discontinuations were as a result of infection. Serious infections included UTI, pneumonia, and bronchitis. There were two reports of opportunistic infection—vulvovaginal candidiasis and pulmonary tuberculosis.

Malignancies were reported in four (1%) patients and included breast cancer, basal cell carcinoma, anal cancer, and a pulmonary carcinoid tumor. Investigators believed the anal cancer and pulmonary carcinoma cases were related to DAC HYP treatment, though there was no observed pattern in malignancies [66]. Cutaneous events occurred in 28% of patients, with the most frequent cutaneous AEs being rash (7%), allergic dermatitis (5%), and eczema (3%). Cutaneous events in 3% of patients led to treatment discontinuation as shown in Table 3. Serious cutaneous events occurred in 2% of patients: urticaria was reported in two patients; and Stevens–Johnson Syndrome (SJS) in one case, which was not validated by a dermatologist and did not meet standard SJS diagnostic criteria. Drug-related hepatic disorders were found in 15% of patients, only 1% of which were serious.

Gastrointestinal AEs were observed in 16% of patients, six (1%) of which were serious inflammatory gastrointestinal events, including ulcerative colitis (three patients), and colitis, Crohn’s disease, and hemorrhagic enterocolitis (one patient each). Overall the safety profile in the SELECTED open-label extension was felt to be comparable to that observed in the SELECT trial, and the risks associated with DAC HYP did not appear to increase with longer durations of therapy [66].

## 7. Adverse Events: DAC HYP Post-Marketing

### 7.1. Cutaneous Events

A prospective open-label study evaluated the risk of development of cutaneous AEs over a 42-month period, as shown in Table 3 [67]. Twenty-three participants (77%) developed new or recurring cutaneous AEs. The majority of cutaneous eruptions presented with localized eczematous and in some cases progressive patches. All moderate to severe rashes (6/23) had psoriaform features, which persisted in later stages. Six patients developed mucosal lesions, the most common of which were recurring aphthous ulcers. Non-eczematous facial rashes included eyelid edema with erythema or diffuse facial erythema. Scalp involvement resembling seborrheic dermatitis was observed in six patients. Eight patients presented cyclic exacerbations of skin symptoms. Skin manifestations developed in most patients within the first 12 months of treatment, and commonly recurred two weeks after DAC HYP injection. There was no correlation between duration of DAC HYP treatment and rash development, and was not associated with MS disease course or with efficacy of DAC HYP therapy in suppressing CNS inflammation. However, a predisposing factor was a prior history of eczema or seborrheic dermatitis.

Biopsies from eight patients showed features of eczematous dermatitis, including scattered CD25^+^ and T_reg_ FOXP3^+^ cells. A consistent finding, most notable in moderate and severe rashes, was a robust population of CD56^+^ cells, but the degree of CD56^+^ infiltration did not correlate with the degree of expansion of this cell population in peripheral blood.

The cyclic pattern of eruptions and/or their temporal relationship to DAC HYP dosing in 8/31 patients (25.8%), including three patients with severe cutaneous AEs, led the authors to conclude that these events may be drug-related, and that by altering normal immune networks, DAC HYP could induce adverse cutaneous reactions resulting in the expansion of CD56^+^ cells, the hallmark of DAC HYP-related skin inflammation. As previously noted, CD56^bright^ NK cells are part of the ILC lineage, and ILC are strategically placed at pathogen entry locations such as skin and mucous membranes. Since DAC typically skews differentiation of common ILC precursors toward CD56^bright^ NK cells, DAC-driven changes in ILC differentiation, while beneficial for CNS inflammatory disease, might in some patients lead paradoxically to enhanced skin reactivity. Because DAC prevents the expansion of T_reg_ cells, this may further contribute to heightened skin reactivity in affected patients [67]. Contrary to most of the literature on drug-induced cutaneous and systemic reactions, which has been attributed to the adaptive immune system, SAE to DAC HYP might suggest a larger role for CD56^bright^ NK cells and the innate immune system in this process [68,69,70]. Further study in this area is warranted.

### 7.2. Drug Reaction with Eosinophilia and Systemic Symptoms (DRESS Syndrome)

The EMA reported on March 6, 2018 that four DAC HYP-treated patients developed exanthema/skin reactions with the involvement of other organs and eosinophilia, and five additional patients developed multi-organ failure that were possibly immune-mediated and later recognized as DRESS syndrome. DRESS is a life-threatening syndrome manifesting with cutaneous lesions and internal organ involvement; it has a 10% mortality rate. The time of onset of DRESS varies, typically from 2 and 12 weeks following drug exposure, though diagnosis may be missed or unnoticed due to its rare and varied presentation [71]. Typical features include eosinophilia, fever, extensive cutaneous lesions, atypical lymphocytosis, lymphadenopathy, hepatic injury, and renal failure. Lung, cardiac, and CNS involvement with meningitis and/or encephalitis may also occur, including brain edema, rhombencephalitis, and high cerebrospinal fluid (CSF) cell count [71,72]. Brain biopsy in one case contained demyelinating lesions showing conspicuous inflammation, and an abundance of T cells, plasma cells, eosinophils and eosinophilic/lymphocytic/plasma cellular meningitis, leading to a diagnosis of DAC-induced DRESS syndrome of the CNS. In marked contrast to typical DRESS syndrome, DAC-induced DRESS appears to occur principally within the CNS. Though CNS DRESS is exceptionally rare and usually presents with a pattern resembling vasculitis, it is likely that five of the patients with CNS complications in the EMA report had CNS DRESS syndrome [72]. Reduced IL-2 consumption due to DAC HYP blockade of IL-2R-HA may increase the opportunity for IL-2 to interact with ILC2 lineage cells, promoting IL-5 secretion and expansion of eosinophils, offering a potential pathway to CNS DRESS in rare cases [73].

### 7.3. Glial Fibrillary Acidic Protein (GFAP)-α Immunoglobulin (IgG)-Associated Encephalitis and Other Severe Encephalopathy Syndromes

Of the 12 patients with severe encephalopathy syndromes cited in the EMA report, one patient had CNS vasculitis, and several cases were reported as anti-N-methyl-D-aspartate (NMDA) receptor encephalitis. One patient was described who developed GFAP-α IgG-associated encephalitis [74]. This patient presented with behavioral and cognitive changes eight months after starting DAC HYP treatment. CSF contained oligoclonal bands and intrathecal immunoglobulin A (IgA) synthesis. Immunostaining showed GFAP-IgG antibodies in CSF. Although anti-GFAP-IgG may occur in isolation, it has also been in the setting of anti-NMDA receptor encephalitis or CNS vasculitis. Although one-third of cases with GFAP-IgG are paraneoplastic, malignancy was not found with this DAC HYP-treated patient.

As a consequence of these 12 cases of severe encephalopathy, four of which had a fatal outcome, DAC HYP was withdrawn from the market in February 2018. Although the precise mechanisms of these disorders are yet to be illuminated, it is reasonable to postulate that inhibition of T_reg_ cell expansion without an adequate concomitant expansion of immunoregulatory CD56^bright^ NK cells or general immune suppression by depletion of activated effector CD25^+^ T cells may have played a role. These latter cases suggest that redirecting IL-2 away from its high-affinity receptor may reduce activation and expansion of pro-inflammatory effector T cells, but in some cases does so at the expense of T_reg_ cells to such an extent that it may result in severe, paradoxical secondary autoimmune attack on multiple organs, including the CNS [74].

### 7.4. DAC: Is There a Future?

Although efficacious in controlled trials, the SAE leading to DAC withdrawal may preclude it from ever returning to clinical use. Even in a head-to-head trial against IFN-β, its therapeutic benefit was modest compared to more recently approved medications for the treatment of RMS [75,76,77,78]. At perhaps the simplest level, given the immunogenicity of the subcutaneous space, would intravenous administration of DAC have been preferable? Would this have reduced the likelihood of SAE?

The frequency of autoimmune cutaneous AE, and rarer but very serious autoimmune hepatitis and CNS events, including CNS DRESS and anti-NMDA encephalitis, reduce the likelihood of DAC returning as a therapeutic agent. Although SAE are also observed with other RMS medications, these can be prevented through the adoption of risk stratification strategies, such as detection and monitoring for John Cunningham virus infection in patients treated with natalizumab, exclusionary co-morbidities in patients receiving fingolimod or ocrelizumab, or frequent monitoring and early detection of side effects with use of alemtuzumab [79,80,81,82]. However, no risk stratification emerged for patients treated with DAC, with the possible exception of a prior history of hepatic disease, psoriasis, or eczema.

The time and expense of additional research to determine biomarkers for risk and efficacy are considerable barriers to pursuing the continued use of DAC. Despite these obstacles, the ability to modify disease activity in RMS, and possibly other autoimmune diseases, via innate immune system expansion should be recognized as a topic of potential scientific and therapeutic importance and thus justify further investigation into the development of biomarkers of risk or modified treatment protocols. If changes in immune cell parameters—the reduction in T_reg_ cell numbers, combined with the modest percentage decrease in CD4^+^ and CD8^+^ T cells and the expansion in CD56^bright^ NK cells—resulting from DAC treatment contribute to autoimmune AE, this data could theoretically be used to develop an index table which might predict risk for developing secondary autoimmune AE. Data obtained from patients in already executed clinical trials and knowledge of their clinical outcomes could be used to develop such an index. However, the calculation of the risk index value in an individual patient would probably require treating them with DAC long enough to obtain steady state T_reg_, CD4^+^/CD8^+^ T cell and CD56^bright^ NK cell levels, which in of itself might place the patient at significant risk of an autoimmune AE. At a more complex and expensive level, transcriptomic and microRNA analyses of still stored serum samples from patients treated with DAC in past clinical trials could reveal valid predictive biomarkers for both efficacy and risk susceptibility.

The current standard of care for multiple sclerosis, at least as it pertains to disease process modifying drugs, relies upon adaptive immune system manipulation. This ranges from immune pathway alterations, immune cell expansion and maturation, reducing oxidative stress, immune cell sequestration and/or migration into the CNS, to outright antibody-directed lymphocyte and monocyte destruction. Unfortunately, the opportunity to expand therapeutic immune modulation into the potentially promising innate system arena has, at least for the present, been stymied by the SAE profile that accompanied DAC use in RMS. Targeting innate immune system expansion, without suppressing IL-2-dependent T_reg_ expansion through activation of IL-2R-IA directly via a CD122 agonist, might be employed as a strategy to expand the CD56^bright^NK cell population. Nevertheless, at least for the present, we do not anticipate that the lessons learned from DAC will positively impact the process of future agent development, and instead may discourage attempts at further development of agents targeting innate immune modulation for RMS. Despite these challenges, developing additional strategies to promote innate immune system expansion should be employed in preclinical models of RMS, with the hope of developing novel therapeutic agents targeting innate immune pathways.

## 8. Conclusions

DAC HYP was efficacious for the treatment of RMS, demonstrating a reduction in relapse rate and disability progression, a reduction in MRI markers of disease activity, and an improvement in patient-reported outcome measures. DAC HYP may reduce autoimmune reactivity by obstructing IL-2 interaction with its high-affinity receptor on lymphoid cells, thus reducing activation and expansion of effector T cell-driven inflammation. However, most of the evidence to date would suggest that the impact of DAC on the innate immune system is largely responsible for its therapeutic efficacy. Unique to RMS therapeutic agents, DAC drives the innate immune system through the expansion of CD56^bright^ NK cells, further reducing T cell-driven auto-reactivity in RMS. Both these actions appear to underlie DAC HYP efficacy in RMS. However, DAC significantly impairs T_reg_ cell expansion, and, possibly, activation-induced T cell apoptosis, both of which depend on IL-2 interacting with its high-affinity receptor on lymphoid lineage cells, and may largely explain the susceptibility of DAC HYP-treated patients to develop the severe secondary autoimmune disorders that resulted in its removal from clinical use in patients with RMS. Hopefully, this will not dissuade future research into the potential utility of innate immune system modulation in the treatment of autoimmune disorders.

## Figures and Tables

**Figure 1 biomedicines-07-00018-f001:**
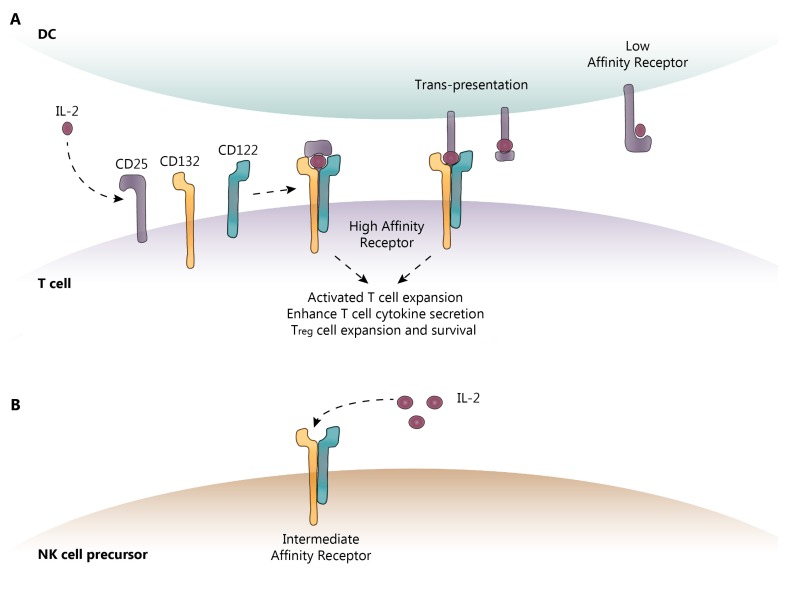
IL-2 receptor (IL-2R) subunits. (**A**) CD25 (α-subunit) is the low-affinity IL-2R and is expressed on activated T cells, T_reg_ cells, and mDC. Once IL-2 is bound to CD25 on the cell surface, CD132 (γ) and CD122 (β) are recruited to form the high-affinity IL-2R. CD132 and CD122 are the only subunits capable of intracellular signaling. (**B**) Other cell types such as NK cells express the intermediate-affinity IL-2R, comprised of CD132 and CD122 only. This receptor requires elevated levels of IL-2 for signaling, such as those found in a local draining lymph node, due to reduced binding affinity for IL-2. DC-dendritic cell; IL-2-interleukin-2; NK-natural killer. Dotted lines show the sequence of events following IL-2 binding to its receptor.

**Figure 2 biomedicines-07-00018-f002:**
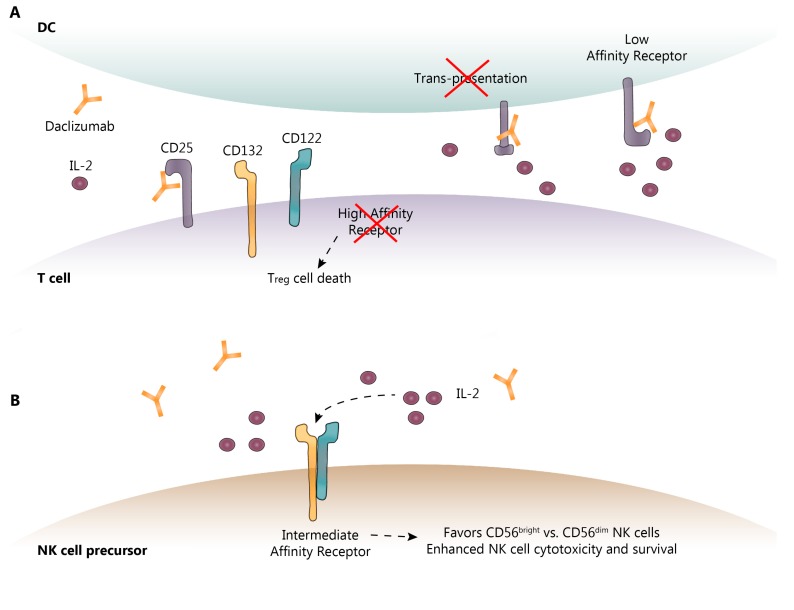
Known and proposed DAC mechanisms of action. (**A**) DAC binds to CD25 blocking its association with CD132 and CD122. This prevents trans-presentation of CD25 from mDC to T cells and ultimately results in effector T cell and T_reg_ cell death; (**B**) As a result of CD25 blockade, serum IL-2 levels are elevated allowing for NK cell activation through the intermediate-affinity receptor, CD132 and CD122, and promoting CD56^bright^ NK cell cytotoxicity and survival. DC-dendritic cell; IL-2-interleukin-2; NK-natural killer. Dotted lines show the sequence of events following IL-2 binding to its receptor.

**Figure 3 biomedicines-07-00018-f003:**
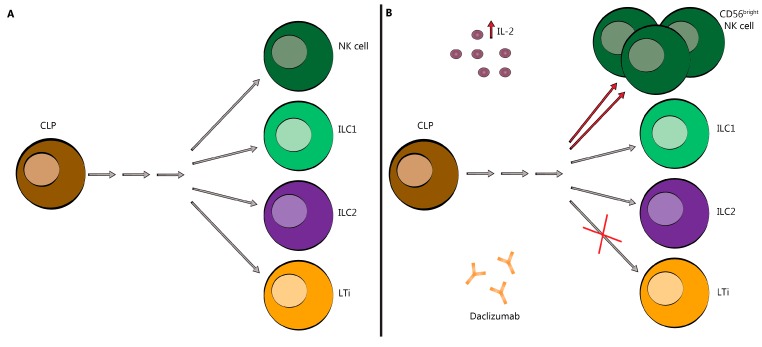
Innate Lymphoid Cells skewed toward CD56^bright^ NK cells following DAC treatment. (**A**) DAC binds to CD25 blocking association with CD132 and CD122. This prevents trans-presentation of CD25 from mDC to T cells and results in effector T cell and T_reg_ cell death. (**B**) As a result of CD25 blockade, serum IL-2 levels are elevated allowing for NK cell activation through the intermediate-affinity receptor, CD132 and CD122, and promoting CD56^bright^ NK cell cytotoxicity and survival (red arrows). It may also redirect the commitment of LTi cells (red ×). CLP-common lymphoid progenitor; NK-natural killer; ILC-innate lymphoid cell; LTi-lymphoid tissue-inducer cell; IL-2-interleukin-2. Grey arrows indicate a series of differentiation steps not depicted in this figure.

**Table 1 biomedicines-07-00018-t001:** Selected Clinical and MRI Outcomes of DAC HYP Treatment in RMS Controlled Trials.

Clinical Trials										
	CHOICE		SELECT			SELECTION			DECIDE	
	DAC ^1^	pl	150 mg	300 mg	pl	Continuous ^2^	Switch ^3^	Washout ^4^	DAC	IFNβ
**Parameters**										
**ARR**	0.27	0.41	0.21	0.23	0.46	0.165	0.179 ****	0.302	0.22	0.39
Risk Reduction, %	34		54 ****	50 ***					45 ***	
**PPRF, %**	52	55	81 ****	80 ***	64	86.4	82.4 ****	75.9	67	51
**CDP, %**										
12 weeks	NR	NR	6*	8	13	7	8 ****	10	16	20
24 weeks	NR	NR	NR	NR	NR	NR	NR	NR	13 *	18
**New Gd+**	1.32 **	4.75	0.3 ****	0.2 ****	1.4	0.2	0.2 ****	0.2	0.4 ***	1.0
Reduction, %	72		79	86		NR	NR	NR	60	
**T2 Lesions**										
New/enlarging	1.1 **	3.4	2.4 ****	1.7 ****	8.1	1.2	2.1 ****	3.3	4.3	9.4
Reduction, %	68		70	79					54 ***	
Volume change, %	ND		−11.1 ****	−12.5 ****	−27.3	−6.9	−8.1 ****	−3.1	0.2 ***	8.6
**Brain Volume, % change**	NR	NR	−0.79	−0.70	−0.74	−0.536	−0.830	−0.551	−0.559 ***	−0.585 ***
**NEDA%**	NR	NR	NR	NR	NR	NR	NR	NR	13 ***	22

DAC—daclizumab; HYP—high-yield process; pl—placebo; RMS—relapsing forms of multiple sclerosis; ARR—annualized relapse rate; CDP—confirmed disability progression; PPRF—proportion of patients relapse-free; Gd^+^—gadolinium-enhancing lesions; IFNβ—interferon beta-1a; NR—not reported; ND—no difference. **^1^** Data reported are for patients receiving high-dose DAC (2 mg/kg); both DAC and placebo received IFNβ; **^2^** Continuous-patients receiving DAC HYP 150 or 300 mg every 4 weeks during SELECT continued their same dose of DAC HYP through additional 52 weeks of SELECTION; **^3^** Switch-patients receiving placebo during SELECT were placed on either 150 or 300 mg DAC HYP for the 52 weeks of SELECTION; **^4^** Washout-patients receiving either 150 or 300 mg of DAC HYP during SELECT were placed on placebo for 20 weeks followed by resumption of their previous dose of DAC HYP administered during SELECT; ********
*p* < 0.0001; *** *p* < 0.001; ** *p* < 0.01; * *p* < 0.05.

**Table 2 biomedicines-07-00018-t002:** Adverse Event Incidence for DAC treated patients in Controlled Trials of DAC HYP.

Clinical Trials							
	SELECT		SELECTION			DECIDE	SELECTED
			Continuous ^1^	Switch ^2^	Washout ^3^		
DAC Dose	150 mg	300 mg	150 mg, 300 mg	150 mg, 300 mg	150 mg, 300 mg	150 mg	150 mg
**AE**							
**Infection (%)**	104 (50)	112 (54)	36 (42),36 (41)	34 (40), 31 (37)	34 (40), 38 (43)	595 (65)	205 (50)
**Serious Infection (%)**	6 (3)	3 (1)	2 (2), 2 (2)	3 (3), 1 (1)	3 (3), 2 (2)	40 (4)	13 (3)
**Hepatic TA (%)**	NR	NR	NR	NR	NR	144 (16)	61 (15)
**AST/ALT:**							
1–3× ULN (%)	54 (26)	62 (30)	30 (35), 30 (34)	23 (27), 22 (26)	21 (24), 26 (30)	NR	NR
3–5× ULN (%)	7 (3)	6 (3)	1 (1), 5 (6)	0, 2 (2)	2 (2), 0	96 (10)	37 (9)
>5× ULN (%)	9 (4)	8 (4)	0, 3 (3)	1 (1), 1 (1)	2 (2), 4 (5)	59 (6)	18 (4)
**Hepatic SAE (%)**	NR	NR	0, 0	0, 0	0, 1 (<1)	6 (1)	5 (1)
**Malignancy (%)**	1 (<1)	2 (<1)	0, 0	0, 1 (1)	0, 0	7 (1)	4 (1)
**Death (%)**	1 (<1)	0	0, 0	0, 0	0, 1 (<1)	1 (<1)	0

DAC—daclizumab; HYP—high-yield process; TA—transaminases; AST—aspartate aminotransferase; ALT—alanine aminotransferase; AE—adverse events; ULN—upper limit of normal; NR—not reported. **^1^** Continuous-patients receiving DAC HYP 150 or 300 mg every 4 weeks during SELECT continued their same dose of DAC HYP through additional 52 weeks of SELECTION; **^2^** Switch-patients receiving placebo during SELECT were placed on either 150 or 300 mg DAC HYP for the 52 weeks of SELECTION; ^3^ Washout-patients receiving either 150 or 300 mg of DAC HYP during SELECT were placed on placebo for 20 weeks followed by resumption of their previous dose of DAC HYP administered during SELECT.

**Table 3 biomedicines-07-00018-t003:** Cutaneous AE Incidence for DAC treated patients in Controlled Trials of DAC HYP.

Clinical Trials								
	SELECT		SELECTION			DECIDE	SELECTED	Post
			Continuous ^1^	Switch ^2^	Washout ^3^			Approval ^4^
DAC Dose	150 mg	300 mg	150 mg, 300 mg	150 mg, 300 mg	150 mg, 300 mg	150 mg	150 mg	
**Cutaneous Events**								
**AE (%)**	38 (18)	45 (22)	15 (17), 21 (24)	17 (20), 11 (13)	19 (22), 16 (18)	344 (37)	114 (28)	23 (77)
**SAE (%)**	2 (<1)	3 (<1)	0, 3 (3)	2 (2), 0	1 (1), 0	14 (2)	8 (2)	6 (19)

DAC-daclizumab; HYP-high-yield process; AE-adverse event; SAE-serious adverse event. **^1^** Continuous-patients receiving DAC HYP 150 or 300 mg every 4 weeks during SELECT continued their same dose of DAC HYP through additional 52 weeks of SELECTION; **^2^** Switch-patients receiving placebo during SELECT were placed on either 150 or 300 mg DAC HYP for the 52 weeks of SELECTION; **^3^** Washout-patients receiving either 150 or 300 mg of DAC HYP during SELECT were placed on placebo for 20 weeks followed by resumption of their previous dose of DAC HYP administered during SELECT; ^4^ Cortese et al., open-label study.

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
