# Peer review of "Daclizumab: Mechanisms of Action, Therapeutic Efficacy, Adverse Events and Its Uncovering the Potential Role of Innate Immune System Recruitment as a Treatment Strategy for Relapsing Multiple Sclerosis"

_biomedicines, 2019, doi:10.3390/biomedicines7010018_

Round 1
Reviewer 1 Report
Overall the manuscript is well drafted.
Please find the attached report for plagiarism check and make appropriate changes such that it is less than 10%.
Specific comments are listed below:
4. Pharmacokinetics and Pharmacodynamics of DAC HYP
Little more details about the exposure-response relations will add value to this article.
Line 175: The t1/2 of elimination was approximately 21 days after the last injection [59].
Question: Is this adults, what is the age range? What is the population which the authors are referring to? Is the PK linear and similar for all genders (male vs female) .
Lines 182-184: “In parallel with increased CD56bright NK cells, serum IL-2 levels increased by approximately 50% within 4 weeks of DAC HYP treatment and Treg cell counts fell by 50-60% within 4 days of a 150 mg DAC HYP injection and returned to baseline levels approximately 20 weeks after the last DAC HYP injection [34,44,46,47].”
Question: What the same dosing used in all the studies which are referenced ? Did the studies show any dose dependent changes for these markers/ end points?
Lines 187: Importantly, there was also a 25% decrease in HLA-DR2-expressing activated effector CD4+ T cells, in DAC HYP treatment in patients with RMS, as further evidence of the down-regulation of pro-inflammatory activity
Question: At what dose is this observed?
Table 1: Formatting. Lines 222-223 are getting mixed with the table 1 explanation.
Conclusions
Can the authors discuss more about the standard of care currently and how learnings from DAC, DAC-HYP aid in streamlining compounds/drug development to treat RMS?

Author Response
1) Overall the manuscript is well drafted. Please find the attached report for plagiarism check and make appropriate changes such that it is less than 10%.
We have examined the manuscript thoroughly and made changes in word choice when possible. However, there are many instances that are noted that cannot be changed, such as: Federal Drug Administration and other organization names; humanized monoclonal antibody, alpha-subunit (CD25), beta-subunit (CD122), activated T cells, annualized relapse rate, five times the upper limit of normal new or enlarging T2 hyper-intense lesions, multiple sclerosis impact scale (MSIS-29), no evidence of disease activity (NEDA), randomized double-blind placebo-controlled trial and other research and clinical terminology/phrasing; ALT-alanine aminotransferase and SAE-serious adverse events and numerous other abbreviations noted in the text, and figure and table captions; Drug Reaction with Eosinophilia and Systemic Symptoms (DRESS syndrome) and other section headings; Providence Multiple Sclerosis Center, Providence Brain and Spine Institute, Portland, OR; High Yield Process; Zynbryta, Biogen, Cambridge, MA and Abbvie, Chicago, IL; central nervous system (CNS); the running header and footer that includes “FOR PEER REVIEW” and “of”; IL-2 receptor (IL-2R) and the IL-2R; the expansion of CD56bright NK cells; etc.
This becomes particularly problematic when summarizing other investigators clinical trials and reporting strictly on clinical outcomes. We have done our best to adjust word choice when possible, to rearrange clinical outcomes when listed in a sentence, etc. We have also been very deliberate to acknowledge the investigators whose trials we are summarizing. We stand by the originality of the manuscript and request that the plagiarism check not include those details that we cannot change, such as those found in addresses, organization names, clinical terminology and phrasing that includes disease names and clinical measuring tools, abbreviations, figure and table captions, and the running header/footer.
2) Pharmacokinetics and Pharmacodynamics of DAC HYP: Little more details about the exposure-response relations will add value to this article.
Line 175: The t1/2 of elimination was approximately 21 days after the last injection [59].
Question: Is this adults, what is the age range? What is the population which the authors are referring to? Is the PK linear and similar for all genders (male vs female).
We apologize for this omission. We have updated the text of the manuscript to include this information, and it reads “Initial studies of DAC HYP in healthy adult subjects (ages 18-65) showed that the maximum serum concentration (Cmax) was 7 days while the t1/2 of elimination was 24 days [59]. In a subsequent study in adults with RMS, Cmax was achieved in 5-7 days and the t1/2 of elimination was approximately 21 days after the last injection [60].” (Lines 173-176)
Lines 182-184: “In parallel with increased CD56bright NK cells, serum IL-2 levels increased by approximately 50% within 4 weeks of DAC HYP treatment and Treg cell counts fell by 50-60% within 4 days of a 150 mg DAC HYP injection and returned to baseline levels approximately 20 weeks after the last DAC HYP injection [34,44,46,47].”
Question: What the same dosing used in all the studies which are referenced? Did the studies show any dose dependent changes for these markers/ end points?
We would like to thank the reviewer for pointing out this omission. The dosing was the same for all the studies, 150 mg or 300 mg every 4 weeks, and the patient population was the same as the OBSERVE, SELECT, SELECTION, and DECIDE clinical trials. For all of the studies, patients were pooled for subsequent analysis, and there was no mention of a difference in any of these parameters between the two DAC HYP doses. We have revised the text to say “For patients treated with 300 mg DAC HYP, the changes in NK cell and Treg frequency were nearly identical; no other results were reported separately for this group [47]. Several other studies examined various immune parameters in pooled DAC HYP-treated patients (150mg and 300mg) from the SELECT, SELECTION, and DECIDE trials. Following DAC HYP cessation, in both 150mg DAC HYP alone and pooled (150mg and 300mg) DAC HYP groups CD56bright NK cells return to baseline pre-treatment levels by 24 weeks after the last injection. In parallel with increased CD56bright NK cells…” (Lines 181-188)
Lines 187: Importantly, there was also a 25% decrease in HLA-DR2-expressing activated effector CD4+ T cells, in DAC HYP treatment in patients with RMS, as further evidence of the down-regulation of pro-inflammatory activity
Question: At what dose is this observed?
We apologize for not including this in the first draft. We have revised the text to include additional detail; it now reads “Importantly, there was also a roughly 25% decrease in HLA-DR2-expressing activated effector CD4+ T cells, in DAC HYP treatment in patients with RMS in both the 150mg and 300mg groups similarly [61]. This coincides with the timing of maximum inflammatory lesion reductions observed in the CHOICE study and provide, as further evidence of the down-regulation of pro-inflammatory activity of DAC HYP [5].” (Lines 192-196)
3) Table 1: Formatting. Lines 222-223 are getting mixed with the table 1 explanation.
We thank the reviewer for noticing this and have fixed the table formatting within the text, and have double checked the remaining tables as well.
4) Conclusions: Can the authors discuss more about the standard of care currently and how learnings from DAC, DAC-HYP aid in streamlining compounds/drug development to treat RMS?
Thank you for the insightful comment. Based on your suggestion and the suggestion of another reviewer, we have included an additional section, “7.4 DAC: Is there a future?”, to address these questions and others about whether there may still be use for DAC as a treatment for RMS or other autoimmune diseases.
Reviewer 2 Report
The authors described in a comprehensive way a very striking story – how the safety issues may result in withdrawal an drug from the market. This is important lesson for medical audience, both clinicians involved in clinical trials and researchers working on mechanisms of drug-induced adverse events. I fully agree with authors that it should not dissuade future research into utility of innate immunity in the treatment of autoimmune disorders.
In my opinion the paper can be published as is, BUT in the section on 7.1 (or 7.2) one may also mention that until now drug-induced cutaneous and systemic reactions (such as DRESS or Stevens-Johnson syndrome) have been usually attributed to adaptive immunity with cytotoxicity involvement [e.g. Zawodniak et al. In vitro detection of cytotoxic T and NK cells in peripheral blood of patients with various drug-induced skin diseases] and evidence of drug-specific immune response [e.g. Porebski In Vitro Assays in Severe Cutaneous Adverse Drug Reactions: Are They Still Research Tools or Diagnostic Tests Already?]. It provides a broader perspective to the readers.
Author Response
The authors described in a comprehensive way a very striking story – how the safety issues may result in withdrawal an drug from the market. This is important lesson for medical audience, both clinicians involved in clinical trials and researchers working on mechanisms of drug-induced adverse events. I fully agree with authors that it should not dissuade future research into utility of innate immunity in the treatment of autoimmune disorders.
In my opinion the paper can be published as is, BUT in the section on 7.1 (or 7.2) one may also mention that until now drug-induced cutaneous and systemic reactions (such as DRESS or Stevens-Johnson syndrome) have been usually attributed to adaptive immunity with cytotoxicity involvement [e.g. Zawodniak et al. In vitro detection of cytotoxic T and NK cells in peripheral blood of patients with various drug-induced skin diseases] and evidence of drug-specific immune response [e.g. Porebski In Vitro Assays in Severe Cutaneous Adverse Drug Reactions: Are They Still Research Tools or Diagnostic Tests Already?]. It provides a broader perspective to the readers.
The reviewer brings up an excellent point. We have added something to this effect in section 7.1 that reads, “Contrary to most of the literature on drug-induced cutaneous and systemic reactions, which has been attributed to the adaptive immune system, SAE to DAC HYP might suggest a larger role for CD56bright NK cells and the innate immune system in this process [68-70]. Further study in this area is warranted.” (Section 7.1, Lines 460-462)
Reviewer 3 Report
The review summarizes current informant that is available on the topic of anti-IL-2 therapies. The research in this filed yielded exciting results that lad to series of relatively successful clinical trials and approval of DAC-HYP anti-IL-2 treatment. Overall, these trials concluded that, although not without adverse reaction, this treatment is well-tolerated and safe. Unexpectedly, after the approval, several patients developed severe reactions that resulted in deaths. Obviously, this forced withdrawal the drag from the shelves.
What went wrong? What lessons can be learned from these experiences?
Can this therapy be salvage and still used in MS or any other diseases?
I think this timely review that addresses some of the above questions.
The manuscript is well-written and exhaustively describes the history of several clinical trials. It provides main study designs and highlights the main results of these trials
I would encourage authors to discuss their opinion about the future of this therapy.
Author Response
The review summarizes current informant that is available on the topic of anti-IL-2 therapies. The research in this filed yielded exciting results that lad to series of relatively successful clinical trials and approval of DAC-HYP anti-IL-2 treatment. Overall, these trials concluded that, although not without adverse reaction, this treatment is well-tolerated and safe. Unexpectedly, after the approval, several patients developed severe reactions that resulted in deaths. Obviously, this forced withdrawal the drag from the shelves.
What went wrong? What lessons can be learned from these experiences? Can this therapy be salvage and still used in MS or any other diseases? I think this timely review that addresses some of the above questions.
The manuscript is well-written and exhaustively describes the history of several clinical trials. It provides main study designs and highlights the main results of these trials. I would encourage authors to discuss their opinion about the future of this therapy.
We would like to thank the reviewer for such thoughtful reflection on our manuscript. Based on your suggestion and the suggestion of another reviewer, we have included an additional section, “7.4 DAC: Is there a future?”, to address these questions and others about whether there may still be use for DAC as a treatment for RMS or other autoimmune diseases.